# Development and validation of context-specific components of obstetric violence: Experiences from the central zone of Tanzania

Theresia J. Masoi[1]*, Stephen M. Kibusi[2], Nathanael Sirili[3], Lilian Teddy Mselle[4]

1 Department of Clinical Nursing, The University of Dodoma, Dodoma, Tanzania, 2 Department of Public Health and Community Nursing, The University of Dodoma, Dodoma Tanzania, 3 Department of Development Studies, School of Public Health and Social Sciences Muhimbili University of Health and Allied Sciences, Dar es Salaam, Tanzania, 4 Department of Clinical Nursing, School of Nursing, Muhimbili University of Health and Allied Sciences, Dar es Salaam, Tanzania

* jtheresia2008@yahoo.com

## Abstract

### Background

Despite the known consequences of obstetric violence, studies have encountered challenges in defining and fully understanding obstetric violence. This difficulty arises from a relative scarcity of research addressing the definition of obstetric violence across various cultures and contexts. As a result, there is a lack of consensus regarding the operational definitions of the components of obstetric violence and variations that may be influenced by geographical and cultural factors.

### Objective

This study describes the process of developing and validating the context specific components of obstetric violence in the Central Zone of Tanzania.

### Methods

An iterative mixed-methods design was used, using the following stages; 1. collecting and analysing qualitative data on context specific components of obstetric violence along with a literature review 2. assessing the content validity with 24 maternal health experts and face validity with 27 postnatal mothers and nine health care providers. Descriptive analysis was employed to analyse participants' characteristics and Likert scale responses from experts, postnatal mothers and health care providers. Item-level Content Validity Index (I-CVI) and Item-face Validity Index (I-FVI) was computed for each component.

### Results

Seven categories of obstetric violence components were identified through this process.These included: physical violence, lack of supportive care and treatment,

**Data availability statement:** All data supporting our work are provided within the manuscript

**Funding:** The author(s) received no specific funding for this work.

**Competing interests:** The authors have declared that no competing interest exist.

**Abbreviations:** ANC, Antenatal Care; BBA, Birth Before Arrival; CHW, Community Health Workers; DRCHCo, District Reproductive and Child Health Coordinator; FGD, Focus group discussion; GA, Gestational age; I-CVI, Item level Content Validity Index; IDIs, In-depth interviews; I-FVI, Item-face Validity Index; KIIs, Key-informant interviews; OV, Obstetric violence; TBA, Traditional Birth Attendants; WHO, World Health Organization

subjugation care, an unfavourable care environment, sexual violence, verbal violence, emotional and psychological violence. In addition, 24 subcategories of obstetric violence were identified. The Item-Level Content Validity Index (I-CVI) ranged from 0.791 to 0.958, while the Item-Face Validity Index (I-FVI) ranged from 0.777 to 0.925.

## Conclusion

The validated components of obstetric violence in Tanzania will contribute to a better understanding of the issue within the Tanzanian context.This in turn, may facilitate a more accurate assessment of the magnitude and impact of obstetric violence while helping to identify key areas for intervention and policy development to promote respectful maternity care.

## Introduction

Obstetric violence is a widespread issue affecting woman while pregnant, laboring and postpartum. It manifests in various forms including physical abuse, verbal violence, coercion or unauthorized medical procedures, breaches of confidentiality, lack of informed consent and abandonment leading to life threatening complications [1,2]. Global estimates of obstetric violence range from 15% to 99% [3] with prevalence in Sub-Saharan Africa reported to be between 44% and 52%.In Tanzania, obstetric violence remains a significant concern [4,5].

In alignment with the United Nations' Sustainable Development Goals (SDGs) 3 (Good Health and Wellbeing) and 5 (Gender Equality), addressing obstetric violence is critical to improving maternity care and upholding women rights [6].Both SDG 3 and SDG 5 emphasize respect for human rights and dignity. Women experiencing obstetric violence often face violations of these principles.This affects both their physical health (SDG 3) and their gender equality (SDG 5).The World Health Organization (WHO) advocates for respectful maternity care and emphasizes the importance of dignity, privacy, confidentiality, protection from harm, informed decision-making and support throughout pregnancy, childbirth and postpartum [1,7].

There is no universal consensus on how to define and measure obstetric violence [8].This is despite ongoing research examining the definition, root causes and methodologies for assessing obstetric violence occurring during pregnancy, childbirth and after childbirth. Cultural and geographic factors shape perceptions of obstetric violence with some women unaware of or even normalizing this pervasive form of violence [9].

Tanzania's national guidelines have outlined the typology of disrespect and abuse in maternity care.These include non-dignified care, abandonment, physical abuse, non-confidential care and non-consented care [10].This definition largely draws from external context that may not fully reflect local realities [8,9,11].Understanding context specific components of obstetric violence is essential for shaping effective policies, interventions and healthcare system improvements [12]. Therefore, this study aimed to develop and validate context specific components of obstetric violence in the Central Zone of Tanzania.

## Materials and methods

### Study context

Tanzania's healthcare system follows a pyramidal structure, as dispensaries and health centres providing fundamental services including basic emergency obstetric care (BEmONC) with regional and national referral hospitals offering specialised maternal services and comprehensive emergency obstetric and newborn care (CEmONC) [13,14].

Maternal health care provision remains a major challenge in developing countries such as Tanzania, resulting in ongoing high rates of maternal mortality [15]. One major risk factor for death in such contexts is delivering a baby at home without skilled medical care [16]. Factors linked to homebirth in Tanzania include a hesitancy to seek care at a birthing facility due to prior individual and community experiences of obstetric violence at the facility level. As a result some women avoid facility birth and may then experience preventable birth complications hence contributing to prevailing high levels of maternal and neonatal mortality [17].

This study was conducted in two regions in the Central Zone of Tanzania from July 2023 to June 2024. The first phase included content validation and took place in Dodoma region Tanzania's capital city with a population of 3,085,625 million people [18,19]. Maternal mortality in this region stood at 417 per 100,000 live births (2018/2020) [20], with 35.5% of women delivering at home. One of the reason cited by women who opted for homebirth included fear of mistreatment at the health facility [21]. Face validation with postnatal mothers and healthcare providers occurred at Sokoine health centre in Singida region [19], where 40% of women give birth at home for similar reasons as cited by women in Dodoma region [22].

### Study approach

An iterative mixed-methods approach [23] was used to develop and validate the context specific components of obstetric violence.This process included the following phases;

(i) conducting primary qualitative research (ii) conducting a literature review (iii) assessing content validity of the context specific components of obstetric violence with experts (iv) translating context specific OV components from English into Kiswahili language (v) assessing the face validity of the context-specific component of OV with targeted users (postnatal mothers and health care providers) and (vi) adjusting items based on the feedback from experts and targeted users [9,24,25]. Fig 1 summarises the development and validation processes.

### Phase I: The qualitative study

The qualitative phase explored components of obstetric violence from postnatal mothers, health care providers and key community informants (i.e., religious leaders, ten-cell leaders who are the local grassroots administrative leaders in Tanzania, traditional birth attendants and male partners) in two regions of the Central Zone of Tanzania.

Three data collection methods were used; in-depth interviews (IDIs), focus group discussions (FGDs) and key informant interviews (KIIs). Twenty four IDIs were conducted with postnatal mothers and four were conducted with religious leaders. In addition, six FGDs were conducted with key informants.These included community members, such as ten-cell leaders, religious leaders, and community health workers. Furthermore, 18 key informant interviews (KII) were conducted with health care providers.

All IDIs, KIIs and FGDs were audio-recorded using a digital voice recorder.The IDI and KIIs lasted between 30 and 50 minutes while the FGDs lasted between 60 and 90 minutes. Interviews were conducted in various locations chosen by the participants.These included participants' homes, offices, and private rooms at health facilities.Each focus group discussion (FGD) consisted of 7–10 individuals and took place either at village executive offices or in a participant's home as coordinated by the ten-cell leaders. Socio-demographic information such as age and education level as well as obstetric characteristics like parity and gravidity were collected initially and documented in the information sheet.This was followed

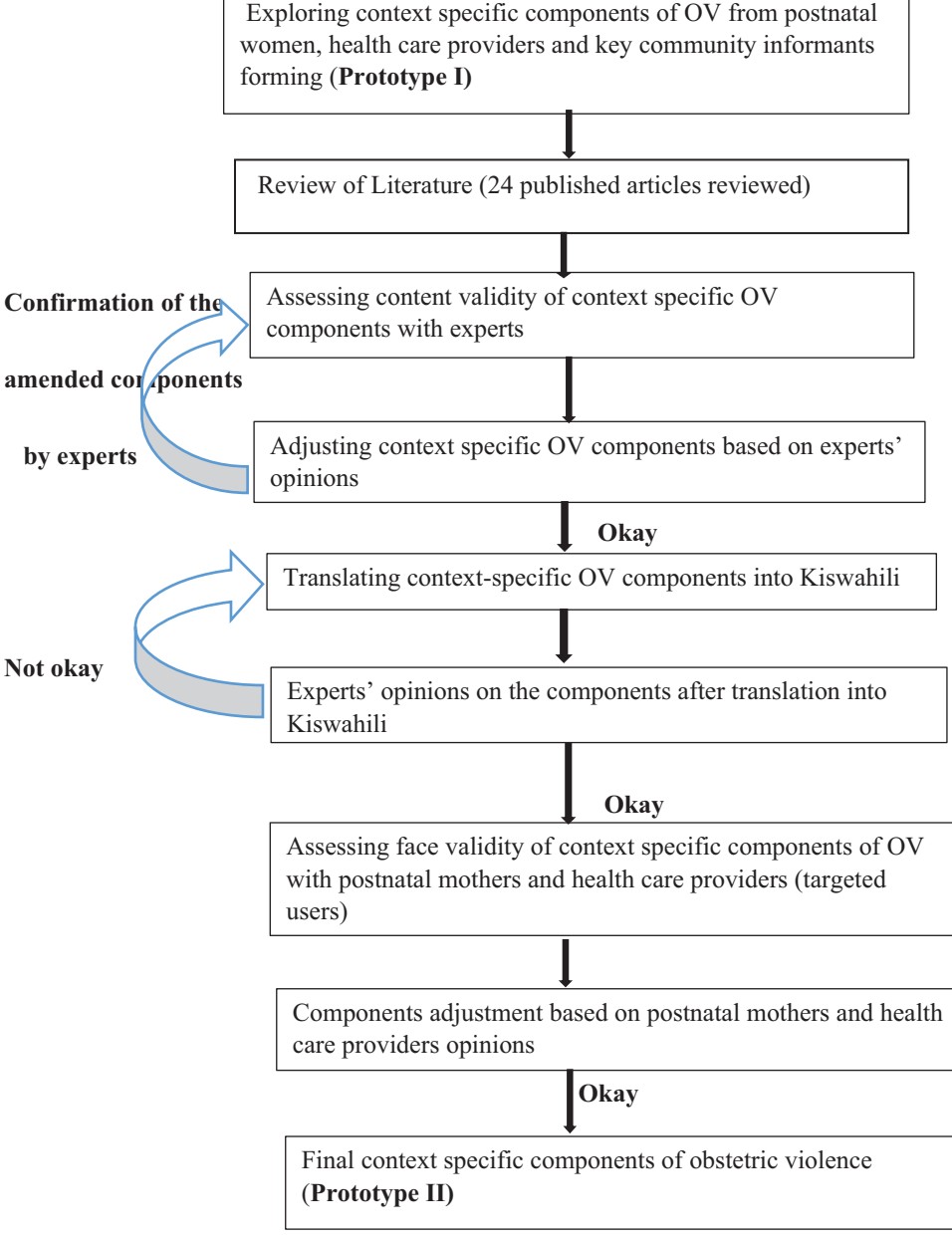

**Fig 1. Development and Validation tree.**

by open-ended questions and probes concerning perspectives on obstetric violence. In order to ensure exploration of comprehensive information about obstetric violence perspectives, the interview guide was kept flexible and was modified during the data collection process.The recorded in-depth interviews (IDIs), key informant interviews (KIIs) and focus group discussions (FGDs) were transcribed verbatim by the researchers and two research assistants familiar with the qualitative research.

Qualitative analysis of the data was performed using a hybrid inductive-deductive approach [26]. Initially, a deductive method guided the development of a preliminary codebook based on the study's objectives and the researchers'

familiarity with the data. An inductive approach then incorporated emerging codes.The authors refined the codebook until a consensus was reached.Data reduction identified meaning units, from which codes were then generated and organised into sub-categories using NVIVO version 12. Similar codes were grouped, and main categories were finalised through comparison and reflection on participants' perspectives.The study identified nine health facility-related and six community-related obstetric violence categories, with 21 and seven sub-categories respectively.These findings formed Prototype 1, which was later validated with experts.

## Phase II: Literature review

The context specific components of OV (Prototype 1) from the qualitative phase were further enhanced by findings from the literature review.The typologies of obstetric violence identified in the review are summarized in Table 1. Literature review findings also served as references during the validation of OV components with experts.This led to the addition of components, such as health system conditions and constraints as suggested by experts during the workshop.

## Phase III: Content validity assessment with experts

**Participants.** Twenty four experts participated in a one day workshop to evaluate the context-specific components of obstetric violence. Studies have recommended a minimum of three to 50 participants for validation processes [25,35,36]. Participants were purposeful recruited based on their expertise and experience in maternal health and/or gender-based violence including obstetric violence. The experts included five obstetricians and gynaecologists from the Association of Gynaecologists and Obstetricians of Tanzania (AGOTA) and local health facilities; 16 nurse-midwives from various local health facilities, representatives from Tanzania Midwives Association (TAMA) and Health Training Institutions (HTIs); two social welfare officers and one expert from the Ministry of Health (MoH). Before the workshop, each participant received an email document outlining the obstetric violence context-specific components.The document defined each category and identified subcategories of obstetric violence, allowing experts to familiarise themselves with the content prior to the workshop. The document also included clear instructions regarding each participant's responsibilities during the workshop.

**The workshop.** A participatory method was utilized during the validation process with the experts.First, the lead researcher oriented participants on the development of Prototype 1 of the context specific components of obstetric violence.Then, participants were asked to independently rate the components of obstetric violence using a Likert scale. The options were: strongly accept, accept, neutral, reject or strongly reject.They also provided justifications for their responses. Some of the criteria used by the experts to rate the components of OV included: relevance, clarity and understandability, cultural sensitivity, applicability in the Tanzanian context and fairness to ensure no bias toward any particular group. Each expert independently reviewed the components and provided written feedback.

After individual assessment, the experts were divided into four teams. Each team was comprised of six members with diverse expertise and institutional affiliations. The teams thoroughly examined the components and provided feedback in plenary discussions. Experts suggested additional or omission of components deemed irrelevant to the Tanzanian context based on their experience.The same criteria used for individual ratings were applied in groups and plenary discussions. During this process, key points were captured through careful note taking of the proceedings. The small group deliberations lasted approximately two hours. At the end, each team presented their conclusions in a plenary session, receiving feedback from other teams until consensus was reached on each category and subcategory.During the plenary discussion, experts also reviewed previous published studies for reference [1,2,23,27,30–33,37]. Prototype I of the context-specific components of OV was then refined based on the experts' consensus and recommendations.

## Phase IV: Translation of the components

The context-specific components of OV validated by experts, was then translated from English to Kiswahili, Tanzania's national language [25]. Two language experts from the Department of Linguistics at the University of Dodoma conducted

**Table 1. Typology of obstetric violence components identified from literature review.**

| SN | Category | Codes/items | References |
|---|---|---|---|
| 1 | Physical abuse | Women beaten, slapped, kicked, or pinched during delivery. Women physically restrained to the bed or gagged during delivery, slapping, shouting, intimidation. Repetitive and aggressive digital vaginal examination | [1,2,9,23,27,28] |
| 2 | Verbal abuse | Harsh or rude language judgmental or accusatory comments, cursing and insulting, a lack of communication, the care providers' avoidance of explaining and clarifying the issues | [28–30] |
| 3 | Sexual abuse | Sexual abuse or rape, realization of digital vaginal examination without gloves; manipulation of genitals brutishly and disrespectfully, touching the body and rectal examination without consent. | [1,9,31] |
| 4 | Failure to meet professional standards of care | Refusal to provide pain relief or give pain medication, painful vaginal exams, performance of unconsented surgical operations, breaches of confidentiality. Unconsented medical procedures (including sterilization), iatrogenic procedures, abusive use of oxytocin, immobilization in the bed during labor, delivery in lithotomy position, routine amniotomy, prolonged fasting without indication, inadequate management of pain without justification, no skin-to-skin contact and early clamping of umbilical cord. | [28,31,32] |
| 5 | Gross violations of privacy | No cubicles, no curtains | [28,31,32] |
| 6 | Neglecting women during childbirth and refusal of admission to health facilities | Negligent care, abandonment, refusal to promote care for women considered "complaining", "scandalous", "unbalanced", "non-cooperative" or "questioning" To delay assisting women in situations of abortion | [28,31,32] |
| 7 | Detention of women and their newborns in health facilities | Detention of women and their newborns in facilities after childbirth due to an inability to pay the bills | [1,17,28,33] |
| 8 | Health system conditions and constraints | Physical condition of facilities, staffing constraints, staffing shortages, supply constraints. Lack of privacy, lack of redress, bribery and extortion, unclear fee structures, unreasonable requests of women by health workers | [1,9,17,28] |
| 9 | Poor rapport between women and providers | Poor communication, dismissal of women's concerns, language and interpretation issues, poor staff attitude | [1,9,23] |
| 10 | Stigma and discrimination | Discrimination based on ethnicity/race/religion, discrimination based on age, discrimination based on socio-economic status, discrimination based on HIV status | [2,17,33,34] |

the translation.Afterward, the content was reviewed by maternal health experts to assess the clarity of the translation. The Kiswahili version of the components was then presented to postnatal mothers and healthcare providers (the targeted users) for face validity testing.

## Phase V: Face validity testing with postnatal mothers and healthcare providers

Face validation of obstetric violence context-specific components was conducted with 27 postnatal mothers during their six week post delivery visits to the health facility.Women were provided with a written tool containing OV components on a Likert scale, with five options ("strongly accept," "accept," "neutral," "reject," and "strongly reject"). They were requested to provide feedback on the clarity of wording, simplicity of the language, understandability, cultural appropriateness and perceived value of each component. Women were also asked to rank the importance of each item for themselves and other women [23,24]. Notes were taken during discussions to capture key feedback.Significant revisions focused on making the language more understandable and ensuring each item's relevance to women. Based on the scoring, all items remained in the final version as women considered them important in the Tanzania context.

In addition, nine healthcare providers (six midwives and three medical doctors) working in maternity units were interviewed.Through face-to-face interviews, each provider rated the OV context-specific components on a Likert scale.

Providers' feedback was sought to improve the applicability of the items and rank their importance in the Tanzanian context. All comments received helped to refine the categories and sub-categories.

## Phase VI. item adjustment

The categories and sub-categories underwent revisions following content validity testing with maternal health experts and face validity assessments with postnatal mothers and healthcare providers in the study area. This involved eliminating certain items, merging items with similar concepts, and adding items deemed necessary based on feedback from experts, postnatal mothers and providers. Final decisions regarding the structure and content of the components **(prototype II)** was reached through consensus among the research team and based on additional review of relevant literature.

## Analysis

This study analysed data using version 29.0 of the Statistical Product for Service Solutions (SPSS) software program. Before analysis, data cleaning procedures were conducted using frequency distribution tables to ensure data completeness and accuracy. Descriptive analysis was employed to examine participants' characteristics, determining the frequencies and percentages of their distribution. Additionally, descriptive statistical analysis was utilized to ascertain the frequencies and percentages of Likert scale responses among participants, ultimately identifying prevalent recommendations in each component. Recommendations with a high prevalence were used in item adjustment.

The criteria for considering a recommendation with high prevalence included the frequency of responses from experts, shared experiences among the experts, the significance of the component within the Tanzanian context and current evidence from other studies and reports referenced during the workshop. The cut-off points were determined based on expert judgment and consensus. A component was considered "high prevalence" if accepted by 75% or more of the experts, "medium prevalence" if accepted by 50%−74%, and "low prevalence" if accepted by less than 50% of the participating experts. This scoring system has also been used by other studies [36].

Moreover, an Item-level Content Validity Index (I-CVI) was computed for each item. The I-CVI was calculated for each item by dividing the number of experts who agreed with the item in terms of its adequacy and its relevancy to the context by the total number of experts involved. This was also among the criteria used for the amendment or removal of the component [38]. Items with an I-CVI value of 0.78 or higher were considered to possess strong content validity as at least 78% of experts agreed that an item was relevant. This threshold aligns with recommendations from the literature [38–40]. For this study the I-CVI values ranged from 0.791 to 0.958, so all the items were considered to be contextually important by all the experts.

In addition, the item face validity index (I-FVI) was calculated by dividing the number of postnatal mothers and health care providers who agreed with the comprehensiveness and clarity of the item by the total number of postnatal mothers and health care providers who participated in the face validation process [41]. An I-FVI value of 0.75 or more signified that at least 75% or more of respondents found the components clear and understandable [38–40,42]. For this study the I-FVI for all components ranged between 0.777 and 0.925.

## Ethics approval and consent to participate

Ethical clearance for this study was obtained from the Senate Research and Publications Committee of Muhimbili University of Health and Allied Sciences in Tanzania (Reference number: DA.282/298/01.C/1758). Permission for data collection was obtained from the Tanzanian Ministry of Health, local government ministry, regional and district authorities, heads of health facilities and community leaders. The standard process for obtaining informed consent was performed. Written informed consent was obtained from each study participant before their participation. The study objectives were explained to study participants and they were encouraged to communicate freely as their information would only be used for research purposes and not affect their care. They were informed that their participation was voluntary and they could withdraw from the interview or discussion any time. Confidentiality of the information was also ensured.

## Results

### Demographic characteristics

**Maternal health experts.** Twenty four experts were involved in the validation processes, including two (8.33%) obstetricians from Association of Gynaecologists and Obstetricians of Tanzania (AGOTA), and three obstetricians (12.5%) from health facilities. In addition, two (8.33%) were midwives from the Tanzania Midwives Association (TAMA) and 10 (41.7%) midwives from health facilities as shown in Table 2. The mean age of the experts was 38.5 years with a range of 25–49 years old.

**Postnatal mothers.** Twenty-seven postnatal mothers, six weeks after delivery were involved in face-to-face interviews as part of the face validity testing. Their mean age was 27 years and the majority 21 (77.8%) ranged between 20 and 34 years of age. Additionally, 13 (48.1%) of postnatal mothers interviewed, had between 2–4 children. Table 3 summarises other demographic characteristics.

### Components of obstetric violence after the validation processes

The first draft (Prototype 1) had nine categories of contextual components of OV that were validated with experts, mothers and providers (targeted users).During the validation process some items were eliminated and merged with similar concepts. New codes and subcategories deemed important based on feedback from experts and targeted users were added. The final draft included the following health facility and community related OV: physical violence, lack of supportive care and treatment, subjugation care, unfavourable care environment, sexual violence, verbal violence, emotional and psychological violence.Three additional sub-categories were added during the validation process with experts from the category of health system condition and constraints and taken from new WHO evidence on mistreatment [43]. Therefore, the final draft (prototype two) has a total of seven obstetric violence components and 24 sub-categories that are linked to the health facilities and community.The final components of OV are presented and summarized in Table 4.

**Physical violence.** The majority of the experts (91.6%) and postnatal mothers (99%) agreed with all the items in the physical violence category. A few additional codes were added, such as pinching with nails, performing of episiotomy and perineal laceration repair without local anaesthesia and forcing a woman to wash blood/faecal stains clothes immediately after delivery.The practice of using hot water to massage the abdomen after delivery was also discussed and deemed harmful due to the risks of burns and severe bleeding.The item-level Content Validity Index (I-CVI) for physical violence was 0.937.

**Lack of supportive care and treatment.** Before the validation process, this category was called "lack of supportive care during pregnancy, childbirth, and after delivery." It was renamed to "lack of supportive care and treatment" to emphasize that treatment also requires supportive measures. New items were added, including refusal of antenatal care (ANC) due to lack of required investigations or coming from a different geographic area and delays in care after home

**Table 2. Experts characteristics.**

| Participants' category | n (%) |
|---|---|
| Obstetricians (AGOTA) | 2 (8.33%) |
| Obstetricians from health facilities | 3 (12.5%) |
| Experts from the Ministry of Health | 1 (4.17%) |
| Midwives from TAMA | 2 (8.33%) |
| Midwives from health facilities | 10 (41.7%) |
| Experts from training institutions | 4 (16.67%) |
| Social workers | 2 (8.3%) |
| **Total** | **24 (100%)** |

**Table 3. Demographic characteristics of postnatal mothers.**

| Variable | Frequency (n) | Percentage (%) |
|---|---|---|
| **Mean age 27 Years** | | |
| **Age (Years)** | | |
| <20 | 1 | 3.7 |
| 20-34 | 21 | 77.8 |
| ≥35 | 5 | 18.5 |
| **Education level** | | |
| No formal education | 1 | 3.7 |
| Primary Education | 20 | 74.1 |
| Secondary Education | 4 | 14.8 |
| College/University level | 2 | 7.4 |
| **Occupation** | | |
| Peasant | 9 | 33.3 |
| Pet trader | 12 | 44.4 |
| Housewife | 6 | 22.2 |
| **Marital Status** | | |
| Single | 2 | 7.4 |
| Married | 19 | 70.4 |
| Cohabiting | 6 | 22.2 |
| **Parity** | | |
| 1 | 6 | 22.2 |
| 2-4 | 13 | 48.1 |
| ≥5 | 8 | 29.6 |

delivery or birth before arrival (BBA).In addition, the component "left unattended without clear information" was added. The Item-level Content Validity Index (I-CVI) for this category was 0.96.

**Subjugation care.** During the plenary discussion with experts, certain items within this category were modified. For example "not allowing time for a woman to pray" was rephrased to "lack of freedom to practice non-harmful religious and cultural practices such as praying." There was disagreement among experts about whether women should be allowed to take the placenta home.Some experts supported this, suggesting that women should be educated about how to handle it, while others raised concern about the risk of infection. A consensus was reached allowing women to take the placenta home, provided they receive education about safe handling. Language modifications were also made to some items and sub-categories for clarity. The Item-level Content Validity Index (I-CVI) for this category was 0.833.

**Unfavourable care environment.** The majority (96%) of experts reported this issue as common in Tanzanian context. Some categories such as lack of information about labor progress, forced treatment and procedure and non-consented care, were merged as sub-categories under this category. In addition, lack of privacy and confidentiality, lack of resources and policies, and facility culture were also incorporated.During the plenary discussion, some items were modified such as changing "examined in presence of so many students" to "examined in presence of other unidentified individuals without consent" as it was noted that the issue was not limited to students only but also included others who should not be present. A new code regarding medication and sample collection without consent was added.The Item-level Content Validity Index (I-CVI) for this category was 0.958.

**Sexual violence.** During the plenary discussion, some experts shared their experience of receiving complaints from women that some of the male providers tend to prolong vaginal examinations unnecessarily.This category was discussed extensively during the plenary session with new items added such as denied sexual intercourse during pregnant and

**Table 4. Final contextual components of obstetric violence after validation.**

| Codes | Sub-categories | Categories |
|---|---|---|
| Slapping, biting, kicking, pinching with forceps, mouth covering to prevent a woman from shouting while pushing, the use of force to separate legs apart when pushing, restraining to the delivery bed both hands and legs with linen or ropes, heavy exercise such as squat and jump for the baby to come down, pinching with nail, forcing a woman to wash her blood/faecal stains clothes after delivery | Violent physical act during labor and delivery | Physical violence |
| Fundal pressure, widening of the birth passage without pain medication, tear repair without local anesthesia, using syringe to pull the nipples, painful and hard abdominal examination, heavy exercise such as squat and jump for the baby to come down, rough per vaginal examination, insufficient pain relief, forceful removal of the placenta, stretching the cervix with hands | Painful routine procedures that lack evidence but are done by experienced providers | |
| Pulling the external genitalia to stimulate labor pain, not giving enough time to rest during pregnancy and after childbirth, use of nail and can shell to widen the birth passage during delivery, massage of the abdomen using hot water after delivery, being hanged on the roof to deliver the retained placenta, doing heavy works like farming, grazing, being beaten, pushed, and hit | Physical violence related to traditional practices during pregnancy, childbirth and after childbirth | |
| Denying services at the antenatal clinics because a pregnant woman is not accompanied by her male partner, denying admission in labor ward due to lack of requirements, denying birth companion of her choice, denying ANC services if a woman is coming from other catchment area, denying preferred birthing position, denying ANC services because a pregnant woman did not perform the required investigations | Denying care and treatment | Lack of supportive care and treatment |
| Delaying care and treatment at health facility after home delivery and birth before arrival (BBA), leaving a client unattended without medical attention and with no clear information, providers being busy with phones and ignoring women request and calls, no pain relief, left in uncomfortable position, giving birth on the floor unattended | Delayed and neglected care and treatment | |
| Not allowing a woman to practice non-harmful social cultural and religious practices such as; praying, taking the placenta home, assuming the birth position of her choice and unjustified restriction of foods and drinks | Loss of control of what is happening during pregnancy, childbirth and after childbirth | Subjugation care |
| Being treated like a child who does not know anything and not being involved in decision making of her treatment and procedures | Infantilization | |
| Repeated vaginal examination for learning purpose, being considered an object upon which providers can act upon, being treated like work objects in the maternity rooms and undervalued as a person | Objectification | |
| Not informed of labor progress and procedures, not clearly informed of caesarean section, episiotomy, repeated painful vaginal examinations and abdominal examination by different professionals without consent, doing procedures and collecting samples without consent | Repeated procedures and treatment without consent | |
| Forced to deliver by operation without proper assessment and justification, forced sterilization by husband after cesarean without woman's consent | Forced treatment and procedure | |
| Being examined in an open space with no curtains or wall partition, being examined in presence of other unidentified individuals without her consent, discussing personal history and treatment in earshot of other women | Lack of privacy and confidentiality | Unfavorable care environment |
| Bribery and extortion, unclear fee structure, unreasonable requests of money and gift by health care providers before care provision, lack of redress | Facility culture and lack of policies | |
| Staffing constraints, physical condition of facilities, supply constraints | Lack of resources | |
| Inappropriate massaging of breast during examination in ANC visits, inappropriate sexual acts from health care providers like clitoral stimulation during vaginal examinations,rape | Inappropriate sexual behaviour by health care providers | Sexual violence |
| Suturing the vaginal opening after delivery to narrow the entrance under husband request for sexual enjoyment | Husband stitches | |
| Denied sexual intercourse during pregnancy, being poorly prepared for sexual intercourse while pregnant, forced sexual activities during pregnancy and after delivery at home, forced sexual intercourse in a position which is not comfortable during pregnancy | Inappropriate sexual practice while pregnant by husband | |

*(Continued)*

**Table 4.** (Continued)

| Codes | Sub-categories | Categories |
|---|---|---|
| Sexual comments regarding body part, shouting, spoken rudely, mocking, unfriendly attitude from providers, threats to gain woman compliance, disrespectful expressions and words, the use of jargons such as "you did not cry when you were doing it" the use of abusive harsh words especially when women fails to push | Poor communication pattern | Verbal violence |
| Unfriendly comments about body during pregnancy from partners and relatives, insults, disrespectful words | Unfriendly language and comments from partner and relatives | |
| Blaming a woman for a bad birth outcome such as losing a baby during delivery, being blamed for being unable to push the baby timely by health care providers as, being blamed for crying during labor, being nicknamed, criticized, being laughed at and labelled a fearing woman, blaming and discriminatory words in situations such as abortion | Blaming a woman for poor child birth outcome | Emotional and Psychological violence |
| Detention of women and their new-born at health facilities because of not clearing the hospital bills related to care during child birth | Detention of women and her new-born to the health facility | |
| Differential treatment based on personal relationship with providers and political position, differential treatment based on health status including women with HIV, treating a woman differently based on her age, parity and marital status, economic and cultural background. | Stigma and discrimination/judgmental care | |
| Being chased away from home because of pregnant or forced to marry | Being chased from home or forced to marry | |
| Rejecting pregnancy responsibilities, being neglected both at home and denied care at health facilities by husband relatives while pregnant or after delivery, abandoned and denied basic needs by husband/close relative while pregnant | Abandoned and denied basic needs by husband/close relative while pregnant | |
| Forced birth companion not of her choice by husband or relatives, given herbal medication for abortion without her willingness, forced to be attended by mother in-law before attending to the health facility for delivery, being forced to go to the village to wait for delivery, forceful insertion of herbal medication to the vagina to initiate labor and hasten the delivery process, being forced to deliver at home by husband or relatives | Coerced reproductive health practices | |

being forced to adopt uncomfortable position by partner. Some postpartum women during face validation expressed feeling coerced into sexual activity with their partners during pregnancy, even when they are not feeling well. Partners may pressure them by claiming that sexual pleasure is enhanced during pregnancy, suggesting that it makes women feel warmer. Both experts and postnatal mothers identified this issue as a significant concern and categorized it as a form of sexual violence during pregnancy as it occurs without the woman's full consent. The Item-level Content Validity Index (I-CVI) in this category was 0.833.

**Verbal violence.** This category was described as the most common type of violence in the Tanzanian context, with the Item-level content validity index of 0.96. Some similar items such as "harsh words" and "abusive words" were merged. Both experts and postnatal mothers agreed with all the items in this category and noted that this type of violence is particularly common during labour and delivery, especially if a woman fails to push the baby in a timely manner while healthcare providers have schedules/timelines in mind that have no correlation to the pregnant person's health or wellbeing. Some postnatal mothers reported that this was the most common reason for why women hesitate to seek care and deliver at health facilities.

## Psychological/emotional violence

Stigma and discrimination were merged into this category with new items added, including the tendency to blame women for delays in seeking health services after the onset of labor pains. Women are often ridiculed and labelled as fearful and may be treated differently based on factors such as age, parity, and marital status. Experts and postnatal mothers also

suggested that "teenagers and women with high parity" should be recognized as particularly vulnerable to violence and discrimination during care provision in healthcare facilities within Tanzanian context.The Item-level Content Validity Index (I-CVI) for this category was 0.854.

## Discussion

Obstetric violence is recognized globally as a serious issue threatening maternal health and well-being but there is no consensus on how best it should be defined based on cultural and geographical differences [23]. This study aimed to develop and validate context specific components of obstetric violence in Tanzania.These components were derived from qualitative interviews with healthcare providers, postnatal mothers and key community informants as well as from a comprehensive literature review. Furthermore, feedback from maternal health experts, health care providers and postnatal mothers during the validation process was systematically incorporated.The validity of the components was confirmed using experts from maternal and child health, social welfare workers and the targeted users (health care providers and postnatal mothers). This process enriched the content and improved the relevancy of each item in harmony with the local community [41].

There was a consensus among experts that the current guidelines on respectful care in Tanzania may not adequately encompass the perspectives of Tanzanians regarding obstetric violence. This limitation could hinder the accurate measurement of obstetric violence as certain contextual factors may be overlooked. Consequently, some incidents may be underreported. This perspective aligns with findings from other studies [44], which suggest that the definitions of respect, disrespect and violence should be contextually specific due to social, cultural and economic differences.

After the validation processes, the finalized context specific components of obstetric violence consisted of the following categories for health facility and community related OV: physical violence, lack of supportive care and treatment, subjugation care, unfavourable care environment, sexual violence, verbal violence, emotional and psychological violence. All the components showed a strong content validity index ranging from 0.791 to 0.958, and were deemed contextually significant by all experts and targeted users.These results align with a study conducted in the West Bank Palestine, where obstetric violence measurement tools were developed and validated. Achieving a content validity index greater than 0.83 [41] indicating that all the components were valid and culturally acceptable.

During the validation processes, certain items sparked more discussion.One such item was the denial of a woman and her relatives the right to take the placenta home.This issue emerged from interviews with postnatal mothers and community members who expressed a desire to have the placenta buried at home viewing it as an integral part of the baby and a significant spiritual element of the childbirth experience [45]. Some experts supported the practice, emphasizing the importance of providing proper information on handling, while others expressed concerns particularly regarding infection risks.Other studies have also noted that keeping the placenta after childbirth is a common practice in certain cultures and not something to be denied when viewed as a meaningful aspect of the childbirth experience extending beyond its physiological role during pregnancy [46,47].

The plenary discussion with experts also highlighted the use of syringes to manipulate flat/inverted nipples, a practice performed by some healthcare providers.This procedure received mixed reactions from both postnatal mothers and experts; some experts and postnatal mothers considered it a form of violence particularly obstetricians while most midwives supported it as the current method for effectively manipulating flat nipples to aid in breastfeeding in this setting. They emphasized that when performed correctly, it can be less painful and is crucial for preventing complications like hypoglycaemia, jaundice and dehydration in newborns due to delayed breastfeeding. Exploring both clinical rationale and women's own personal experiences is significant in informing a more balanced understanding of how such procedures are interpreted, normalized, or challenged within various stakeholder groups.Other studies have found that the use of the inverted syringe technique in women with inverted nipples is not associated with improvement in breastfeeding rate and associated with complications such as nipple pain, cracks, mastitis and recommend to further investigate other useful methods for manipulating the inverted nipples rather than syringes [48,49].

During the plenary discussion, experts addressed the issue of physically restraining women to the delivery bed using linen or ropes, categorizing it as physical violence.There was a particular concern about managing mentally distressed women during labor, considering staffing constraints. Some experts acknowledged that in certain circumstances such measures may benefit the mother and her pregnancy.This finding is also reflected in a study conducted in Nigeria where the majority of healthcare providers believed that it was only justified to physically restrain a woman if she was experiencing an eclamptic seizure or if she was uncooperative.This view point was not supported by most women who believed that physical restraint was unnecessary in any situation and they perceived it as a display of healthcare providers lacking empathy [50]. Experts emphasized the need for caution and stressed the importance of providing clear information to the relatives involved if there will be no other options than restraining.

The issue of sexual violence, particularly rape, was extensively discussed during the plenary session. Initially, there was a proposal to exclude rape from obstetric violence, as some experts argued that the act of rape, regardless of context, constitutes a form of sexual violence that transcends the scope of healthcare mistreatment.They believed that labeling rape as obstetric violence might suggest it is simply an improper medical practice, rather than a criminal act that should be addressed through legal and judicial channels. However, after reviewing various legal frameworks, consulting with legal professionals within the Tanzanian context and conducting a thorough literature review, all authors ultimately agreed to retain rape as a form of obstetric violence within the category of sexual violence.This decision was based on documented cases of rape occurring in healthcare settings and the inclusion of rape in several studies as a form of sexual violence during obstetric care, particularly when it involves non-consensual sexual acts against pregnant women whether in community or healthcare settings [51–53].

Experts also recommended the inclusion of the category "health system conditions and constraints" that was derived from the World Health Organization (WHO) new evidence on mistreatment of women during childbirth as a category of obstetric violence (OV) and a contributing factor [43]. It was suggested to be added in the category of unfavorable care environment and includes factors such as understaffing, inadequate resources such as insufficient medical equipment or medications that can result in delays in care, insufficient monitoring or hurried procedures potentially compromising the safety and dignity of women during pregnancy and childbirth. Furthermore, institutional policies and practices that prioritize efficiency over patient centered care such as rigid protocols that do not consider individual needs or practices that overlook informed consent can leave women feeling violated. Additionally, inadequate communication between healthcare providers and clients due to language barriers, lack of trained interpreters or insufficient explanations of procedures and treatments can lead to confusion, fear and a sense of violation.Issues like bribery, extortion, unclear fee structures and unreasonable demands for money or gifts from healthcare providers before providing care were also highlighted as concerns and need to be included in all aspects of obstetric violence.

## Strength and limitations

This is one of the first studies in Tanzania to develop validated evidence based context specific components of obstetric violence using a participatory approach that engaged key stakeholders in maternal and child health.The validated framework of obstetric violence can help standardize monitoring of OV incidences and promoting the well being and dignity of women.Ensuring that obstetric practices align with best practices in maternal health thus foster an environment where women's rights and needs are respected.This prototype serves as a valuable training resource for current and future Tanzanian healthcare professionals, to enhance their capacity to identify and prevent obstetric violence.This approach is crucial for improving maternal and neonatal outcomes.

A significant limitation of this study is that, while the prototype identifies and addresses common forms of obstetric violence, it may not fully capture the unique experiences of all individuals in the community. Obstetric violence is a multifaceted issue influenced by diverse cultural, social and personal factors. Furthermore, institutional differences in healthcare practices may influence how obstetric violence is experienced and recognized. However, involving key stakeholders

including women and health care providers in validation process may have mitigated this limitation and as our process involved a multi-disciplinary team of experts in social sciences, midwifery, obstetrics, public health and postnatal mothers who all contributed to refining the prototype. While the prototype provides a valuable foundation for identifying and addressing OV incidences, further adaptations and continuous updates will be necessary to ensure it remains inclusive and responsive to diverse perspectives and experiences.

## Conclusion

The validated obstetric violence components include seven main categories of obstetric violence related to health facilities and community in the Tanzanian context.The components achieved acceptable validity testing.Thus, they are valid context specific components that can be used to develop obstetric violence measurement tools and health education materials to prevent and address obstetric violence in Tanzania.The validated obstetric violence components can aid significantly in addressing this concern.The typologies may assist healthcare providers in recognizing harmful practices and adopting more ethical and compassionate care approaches thereby enhancing the overall quality of care delivered to pregnant women.

The validated components of obstetric violence can also help to empower pregnant women, their families and communities by educating them about their rights and promoting understanding about what practices constitute obstetric violence and how to report or resist harmful practices.The validated components of obstetric violence may be integrated into training initiatives for current and future healthcare professionals to ensure they recognize the different types of obstetric violence. It is essential to provide training that focuses on identifying these behaviours, understanding their effects on women and learning prevention strategies.

Maternal mortality may sometimes be exacerbated by a lack of trust between pregnant women and healthcare providers, especially when women fear violence.By using the validated prototype to foster environments of mutual respect and safety, pregnant women may feel more comfortable seeking care, sharing concerns and adhering to medical advice. This may also reduce the risk of complications and death for women who otherwise would avoid facility birth. Additionally, many forms of obstetric violence, such as neglect can directly contribute to poor maternal and neonatal outcomes including maternal mortality and fetal death.By identifying these behaviours and preventing them, healthcare providers can improve the quality of care potentially reducing avoidable deaths.

We recommend future research to concentrate on identifying the most effective methods for reporting incidences of obstetric violence.Future researchers should also prioritize the identification and evaluation of evidence-based interventions aimed at preventing such occurrences both within healthcare facilities and in community settings. It is essential that future studies work toward developing valid and reliable measurement tools that can accurately assess the prevalence of obstetric violence (OV) and its effects on maternal health as well as to the health of new-borns. It is anticipated that these validated components will be further enhanced as additional studies are conducted in various cultural and geographic contexts thereby contributing to a more comprehensive understanding of obstetric violence and its implications.

## Acknowledgments

Heartfelt thanks to the University of Dodoma for providing some financial support to conduct this study and all the research participants without forgetting Ms. Elisa Brettler Vandervort for her assistance with English grammar editing.

## Author contributions

**Conceptualization:** Theresia John Masoi, Stephen M. Kibusi, Nathanael Sirili, Lilian Teddy Mselle.

**Data curation:** Theresia John Masoi.

**Formal analysis:** Theresia John Masoi.

**Funding acquisition:** Theresia John Masoi.

**Investigation:** Theresia John Masoi.

**Methodology:** Theresia John Masoi, Stephen M. Kibusi, Nathanael Sirili, Lilian Teddy Mselle.

**Resources:** Theresia John Masoi.

**Supervision:** Stephen M. Kibusi, Nathanael Sirili, Lilian Teddy Mselle.

**Validation:** Theresia John Masoi, Stephen M. Kibusi, Nathanael Sirili, Lilian Teddy Mselle.

**Writing – original draft:** Theresia John Masoi, Lilian Teddy Mselle.

**Writing – review & editing:** Stephen M. Kibusi, Nathanael Sirili, Lilian Teddy Mselle.

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
