## [Decision Letter · Decision Letter 0]

PONE-D-24-51014Development and validation of context-specific components of obstetric violence: Experiences from Central Zone TanzaniaPLOS ONE

Dear Dr. Masoi,

Thank you for submitting your manuscript to PLOS ONE. After careful consideration, we feel that it has merit but does not fully meet PLOS ONE’s publication criteria as it currently stands. Therefore, we invite you to submit a revised version of the manuscript that addresses the points raised during the review process. Please submit your revised manuscript by Mar 22 2025 11:59PM. If you will need more time than this to complete your revisions, please reply to this message or contact the journal office at plosone@plos.org. Please include the following items when submitting your revised manuscript:

We look forward to receiving your revised manuscript.

Kind regards,

Florian Fischer

Academic Editor

PLOS ONE

Journal Requirements:

3. We note that your Data Availability Statement is currently as follows: [All relevant data are within the manuscript and its supporting files]

Reviewers' comments:

Reviewer's Responses to Questions

**Comments to the Author**

1. Is the manuscript technically sound, and do the data support the conclusions?

Reviewer #1: Yes

Reviewer #2: Yes

2. Has the statistical analysis been performed appropriately and rigorously? 

Reviewer #1: Yes

Reviewer #2: Yes

3. Have the authors made all data underlying the findings in their manuscript fully available?

Reviewer #1: Yes

Reviewer #2: Yes

4. Is the manuscript presented in an intelligible fashion and written in standard English?

Reviewer #1: Yes

Reviewer #2: Yes

5. Review Comments to the Author

Reviewer #1: This important study investigates components of obstetric violence (OV) in the Central Zone of Tanzania using a mixed-methods approach with a variety of participants, including postnatal mothers, healthcare providers and community members, ultimately identifying seven categories of OV, with acceptable levels of item-level content validity and item-face validity. The manuscript suffers from some distracting grammatical errors and could benefit from further conceptualization of the practical implications of the tool, but with some fine-tuning, this work could contribute importantly to the growing literature on OV.

A few grammatical issues to be addressed throughout the paper: capitalizations (many words are unnecessarily capitalized), comma placement/usage (there are commas throughout the manuscript with multiple spaces before and after them and places where commas are missing where needed, particularly for embedded phrases), lack of periods at the end of sentences, subject/verb agreement. Additional edits suggested below.

Abstract:

Methods:

• The sentence in methods beginning with “The design involved...” is confusing, please clarify. How does the literature review fit into this picture? Consider removing this sentence and listing out the steps of the study sequentially.

• Edit: “they included ‘a’ gynecologist or gynecologist’s’; an obstetrician or obstetrician’s’.

• The term “end users” is confusing here. Please clarify. How will this be “used”?

Results:

• Please remove capitalization of “Unfavorable.”

Conclusion:

• This sentence is confusing and the verb tenses don’t line up. It may also be a fragment. Please re-arrange with concise and strong sentence.

Introduction:

• P. 3, first sentence: consider editing to: “while pregnant, during labor, and after childbirth.”

• P. 3, second paragraph, last phrase: please explain what you mean by, “should be analysed from a different context.” (or remove that phrase)

• Please clarify that these are the UN’s SDGs, then relate these goals back to obstetric violence.

• P. 3, 3rd paragraph: given that obstetric violence is a relatively new concept, I would recommend removing the phrase, “While the forms and underlying causes of obstetric violence are now extensively recorded.” Also consider combining this paragraph with the next one, which could be shortened to just its final sentence.

• P. 4, 2nd paragraph, first sentence starting with “In Tanzania,” this sentence is difficult to follow, consider breaking it up. Same for the sentence beginning with “Without acknowledging…”—it is unclear how this could lead to over-reporting.

Material and Methods:

• P. 4-5: Please add period after end of first paragraph.

• P. 5: First full paragraph: Please either add commas around “Tanzania being one of them,” or rearrange the first sentence. Also, consider providing additional support to the last sentence of this paragraph. This is a very important point, consider adding at least one more sentence that lays out the connection between obstetric violence and maternal mortality. Also please add a period at the end of this paragraph.

• Please explain the process for going from focus group and interviews into the creation of the prototype. What kind of analysis was done? By whom?

Phase III

Participants:

• P. 10- Please remove from first sentence: “if they are contextually valid.”

• P. 10- Consider adding the numbers of participants for each category of expert.

The workshop:

• P. 10- Please clarify the criteria the experts are using to rate the components. Were they rating whether or not these components should be included in the model? Or if these factors are important to their patients? Or which items they’ve had experience with? Please list the specific prompt.

• P. 10- Please clarify how the experts provided feedback on the applicability of the component to the Tanzania context. Was this open-ended feedback?

Phase V: Face validity testing

• P. 11- Same comment as above—please include the prompt the women were responding to on the Likert scale.

Analysis

• P. 13- Please clarify your criteria for considering recommendations with a “high prevalence.” How did you determine the cut-off for high-medium-low?

• P. 13- Please justify your use of 0.78 as your cut-off for the I-CVI and a cut-off of 0.75 for I-FVI.

Results:

Maternal Health Experts:

• P. 14- The age group category in Table 2 is confusing as presented. Is there something you are trying to demonstrate with this variable? If this is meant to show that some of these groups are younger than others, there may be a more effective way to show that. If it is just to provide context, average age may be preferrable.

Physical violence:

• P. 21- Please provide further explanation for “Kicking was reported as impractical.” Were these items assessed for practicality?

Lack of supportive care and treatment:

• P. 21- Please spell out what the acronym ANC stands for the first time it appears in the text.

Sexual violence:

• Please provide further support for removing the category of rape. Provide additional support for this sentence: “The discussion led to the consensus that rape is not just an act of violence but a criminal offense, even though there have been reports of such actions by healthcare providers.” How did rape’s status as a crime fit into the discussion? What were the primary reasons laid out by the group for this item’s removal?

• P. 23- Please clarify the sentence that begins with the phrase, “Some postnatal women…” Does this mean postnatal women suggested adding a new category: “to be forced to have sex with their partner during pregnant even when they are not feeling well”? Please explain, “claiming they are warm.”

Psychological/emotional violence:

• P. 23- Please add the word “differently” to the following item: “treating a woman (differently) based on her age, parity and marital status…”

• P. 23- Please clarify this phrase, “experts and postnatal mothers added teenagers and high parity women as they are so vulnerable to violence during care provision at the health facilities in our settings.” Does this mean additional items based specifically on age and parity were added to the prototype?

Discussion

• P. 26- Please clarify the sentence beginning with, “Experts also recommended the inclusion Health…” It is difficult to follow.

Strength and limitation

• P. 27- Add a strength that highlights the practical utility of this OV prototype—what can it do, how can it advance the field?

• P. 27- Please clarify this limitation. Is there any reason to assume that male partners and/or religious leaders will have more to say on OV than your current participants?

• Consider including a limitation that addresses the difficulty of including all perspectives in one prototype for OV. How did you resolve scenarios of disagreement among your participants? How did you ensure that all of the voices in the focus groups were heard? Is it possible that relations of power played out in the focus groups, such that some voices were louder and therefore had outsized impacts on the resulting model?

Conclusion

• P. 27- Please lay out some of the more specific practical implications of your study—how can this be used to help pregnant people? To help healthcare providers? To address maternal mortality?

• Please lay out the next steps for researching this topic—what are the important questions that remain about OV in this context? What are the important areas for future research?

Reviewer #2: This peer review has been finalized based on the following sub-topics

1. Originality of the study and the results

The study clearly presents the topic which in my view aligns well with a clear goal, methodology, results, limitations and conclusions. I also noted that the study has a good introduction. However, my suggestion is that the authors can consider enriching this introduction with global, regional and national statistics and rates to show the burden of obstetric violence and hence contributing to a clear case for action.

2. Results reported have not been published elsewhere

I have done a search for the topic but I have not found any record of the results having been published anywhere. This is a clear reflection of originality of the results.

3. Experiments, statistics, and other analyses are performed to a high technical standard and are described in sufficient detail.

The goal of developing contextual components of obstetric violence in Central Tanzania was achieved by coming up with clear conceptualization as indicated by Figure 1 which demonstrates prototype 1 and 2. I take note that the iterative-mixed methods that was utilized in the development and validation of the contextual components of obstetric violence is well described in sufficient details. Prototype 1 which involved the review of related literature to identify the typology of obstetric violence from literature is intelligently done. The identified components of obstetric violence were then validated by experts through a plenary discussion based on their applicability to the Tanzanian context. Experts’ additional components of obstetric violence-based on the WHO evidence on mistreatment were included. Prototype 1 components that we validated by experts were then translated from English to Kiswahili by linguistics experts, and validation from the University of Dodoma before the face validity testing with post-natal mothers and healthcare providers. Items also underwent adjustments – a process that involved elimination of some items, merging others with similar concepts and adding others which were deemed relevant to the Tanzanian context.

Descriptive statistics that were used to summarize the experts, and demographic characteristics of postnatal mothers who participated in the study were appropriate and clearly presented in tabular form. This approach was used for data cleaning to assess data completeness and accuracy prior to qualitative analysis. The qualitative analysis of the data that was collected from experts and postnatal mothers was clearly coded, then the related codes were organized into subcategories and then sub-categories were merged to form categories which constituted items.

It is also clear that the authors used descriptive analysis to arrive at frequencies and percentage of the responses of the Likert scales based on the participants’ responses. This approach was used for identification of most common recommendation in each component. High frequency recommendations were utilized in items adjustments. Each of the items were subjected to Item-level Content Validity Index computation which was arrived by dividing the total number of experts who agreed with items in terms of adequacy and its contextual relevance by the total number of experts involved. Authors are clear that they used this method as one of the criteria for item amendment. Items utilized an I-CVI value of 0.78 or higher as a consideration for content validity. It is clear that researchers arrived at a content validity values of 0.791 and 0.958.

Item face validity index (I-FVI) was computed by diving the total number of postnatal mothers and healthcare workers who agreed with the comprehensiveness and clarity of items by the total number of postnatal mothers and healthcare workers who participated in the face validation process. An I-FVI value of 0.75 or more was considered a good score and provided a good into how postnatal mothers and healthcare providers interpreted and responded to the components. For this study the I-FVI ranged between 0.77 and 0.925.

Based on the above analysis seven contextual components of OV were arrived namely: physical violence (I-CVI= 0.937), lack of supportive care and treatment (I-CVI=0.96), subjugation care (I-CVI=0.833), unfavorable care environment(I-CVI=0.958), sexual violence(I-CVI=0.833), verbal violence (I-CVI=0.833), emotional and psychological violence (I-CVI=0.96).

Conclusion: I agree that experiments, statistics, and other analyses are performed to a high technical standard and are described in sufficient detail.

4. Conclusions are presented in an appropriate fashion and are supported by data.

The study concludes that, the validated components of obstetric violence components consisted of seven main categories that were related to health facilities and the community. The conclusion also indicated that the components achieved acceptable validity testing, hence arriving at the conclusion that the seven components were the valid contextual components that can be utilized in development of obstetric violence measurement tools in Tanzania. Authors also recommended further strengthening of the seven components by further studies.

Conclusion: I agree that the conclusion is presented in appropriate version, and are supported by data.

5. The article is presented in an intelligible fashion and is written in standard English.

I agree with this statement. The article is written and clear standard English. However, there are some minor grammatical and punctuation issues which I suggest to the authors to consider correcting. These suggestions are as follows:

Abstract

Results: Seven categories of obstetric violence components were identified, including physical violence, lack of supportive care and treatment, subjugation care, Unfavorable care environment, sexual violence, verbal violence, emotional and psychological violence. In addition, 24 subcategories of obstetric violence were also identified .

Issue: I suggest the removal of the space between identified and the full-stop.

Funding details: The First author received some funds during proposal development in support of her doctorial studies .

Issue: The word doctoral is misspelled. There is a need to remove the space between the word studies and the full-stop.

Ethics Statement: Permission for data collection was obtained from the Ministry of health, Ministry responsible for local government, regional and district authorities, Heads of health facilities and community leaders.

Issue: The word health under the Ministry of health to be capitalized as this represents an institution.

Fields research: … informed that their participation was purely voluntary.

Issue: A full-stop is needed after the word, “voluntary”.

Results: … 24 subcategories of obstetric violence were also identified .

Issue: Same comment as previous.

Introduction: Obstetric violence manifests in various forms, such as physical violence, severe embarrassment, verbal violence, coercion or unauthorized medical procedures like sterilization, breaches of confidentiality, lack of informed consent, and abandonment leading to life-threatening and preventable complications. (1) .

Issue: The first full stop needs to be removed.

Without acknowledging local expressions and interpretations of obstetric violence, there is a risk of underreporting or over-reporting the incidences prohibiting effective efforts to address it (8).

Issue: Kindly check whether the word “prohibiting” is correctly used in this context.

Study Context: From the Health centres, the district hospitals follow then regional referral

hospitals, Zonal referral hospitals and National referral hospitals with more specialized maternal services (9)

Issues: Spacing needed between the words “From the…”, full stop needed at the end of the sentence.

Maternal health care provision remains a major challenge in developing countries Tanzania being one of them leading to prevailing high maternal mortality (10) .

Issues: kindly check spacing between citation number 10 and the full-stop.

‘…community experiences of obstetric violence (12)’

Issues: Full stop needed after citation number 12.

The first phase of content validation process with experts took place in Dodoma (13).

Issues: Kindly check citation 13. I think it is misplaced.

Dodoma a capital City of Tanzania with a population of 3,085,625 million people (14).

Issues: The sentence needs restructuring for clarity.

Postnatal Mothers: Table 3 summarises other demographic characteristics

Issues: Full stop needed after the word “characteristics”.

Components of Obstetric violence after the validation processes

Final draft consist of the following health facility and community related OV;…

Issue: Colon (:) needed after the word “following”

The final components of OV are presented and summarized in table 4 .

Issue: Space between 4 and the full stop to be removed.

Table 4: Final contextual components of obstetric violence after validation

-Being chased from home or forced to be married forced

Issue: The second word, “forced” to be deleted.

Lack of supportive care and treatment

… health facility after home delivery and birth before arrival (BBA) .

Issue: Space between (BBA) and the full stop to be removed.

Some postnatal women also agreed to be forced to have sex with their partner during pregnant even when they are not feeling well, claiming that they are warm.

Issue: Space between BBA and the full-stop to be removed. An s to be added to the word partner. The word ‘pregnant’ to be replaced with the word ‘pregnancy’.

Psychological/emotional violence

New item were added such as blaming a woman for delaying to seek health services after labour pain start

Issue: an s to be added at the end of the word item.

Discussion

… there is no consensus on how best should be defined due to cultural and geographical differences (18).

Issue: the word it to precede the word ‘should’.

… enriching the content and improving the relevancy of each items in harmony with the local community (35).

Issue: s to be removed from the word items to read item.

This view was also reported in other study (38).

Issue: The word ‘study’ to be replaced with the word ‘studies’.

extending beyond its physiological role during pregnancy (40,41) .

Issue: spacing between parenthesis and the full-stop to be removed.

… investigating other useful methods for manipulating the inverted nipples rather than syringes (42,43)

Issue: Full-stop needed after parenthesis.

‘… recommended the inclusion Health system conditions and constraints …’

Issue: the word ‘of’ to be inserted between ‘inclusion’ and ‘health’.

Strength and imitation

Issue: An l to be added to the word imitation to read limitations. The who statement to replaced with the title, ‘Strengths and limitations

This study is one of the first study in Tanzania …

Issue: The word ‘study’ to be replaced with the word ‘studies.

Conclusion

… strengthened as further studies are conducted in other context

Issue: The word ‘context’ to be replaced by the word ‘contexts’. A full stop to be added after the word contexts,

Conclusion – I agree that the article is presented in an intelligible fashion and is written in standard English. However, I suggest that the authors consider correcting minor errors that I have highlighted above.

6. The research meets all applicable standards for the ethics of experimentation and research integrity.

I agree with this statement. Researchers proved that all applicable standards for ethics and of experiments and research integrity were met by the researchers.

7. The article adheres to appropriate reporting guidelines and community standards for data availability.

I agree that the article meets appropriate reporting guidelines and community standards for data availability.

6. PLOS authors have the option to publish the peer review history of their article (what does this mean?). If published, this will include your full peer review and any attached files.

Reviewer #1: No

Reviewer #2: **Yes: **Melchizedek Nyakundi Mokaya

---

## [Author Response · Author response to Decision Letter 1]

20 Mar 2025

Reviewer #1: Thank you for reviewing our manuscript and for the constructive comments

1.This important study investigates components of obstetric violence (OV) in the Central Zone of Tanzania using a mixed-methods approach with a variety of participants, including postnatal mothers, healthcare providers and community members, ultimately identifying seven categories of OV, with acceptable levels of item-level content validity and item-face validity.

Response: We appreciate this recognition

2. The manuscript suffers from some distracting grammatical errors and could benefit from further conceptualization of the practical implications of the tool, but with some fine-tuning ; this work could contribute importantly to the growing literature on OV. A few grammatical issues to be addressed throughout the paper: capitalizations (many words are unnecessarily capitalized), comma placement/usage (there are commas throughout the manuscript with multiple spaces before and after them and places where commas are missing where needed, particularly for embedded phrases

Response: The whole document has been edited for grammar and spelling errors

3 Abstract

Methods:

•The sentence in methods beginning with “The design involved...” is confusing, please clarify. How does the literature review fit into this picture? Consider removing this sentence and listing out the steps of the study sequentially. Edit: “they included ‘a’ gynecologist or gynecologist’s’; an obstetrician or obstetrician’s’.

•The term “end users” is confusing here. Please clarify. How will this be “

Response:

• The sentence has been revised and the steps of the study have been sequentially written and explained in detail in the methods section of the manuscript page 5-13

• The term end user ,has also been edited to targeted users referring to pregnant/postnatal women and health care providers

Results:

Please remove capitalization of “Unfavorable

Response: The capitalization have been removed

Conclusion:

•This sentence is confusing and the verb tenses don’t line up. It may also be a fragment. Please re-arrange with concise and strong sentence.

Response: The whole paragraph has been re-written .Page 2-3

4 .Introduction

• P. 3, first sentence: consider editing to: “while pregnant, during labor, and after childbirth.”

• P. 3, second paragraph, last phrase: please explain what you mean by, “should be analysed from a different context.” (or remove that phrase)

• Please clarify that these are the UN’s SDGs, then relate these goals back to obstetric violence.

• P. 3, 3rd paragraph: given that obstetric violence is a relatively new concept, I would recommend removing the phrase, “While the forms and underlying causes of obstetric violence are now extensively recorded.” Also consider combining this paragraph with the next one, which could be shortened to just its final sentence.

• P. 4 , 2nd paragraph, first sentence starting with “In Tanzania,” this sentence is difficult to follow, consider breaking it up. Same for the sentence beginning with “Without acknowledging…”—it is unclear how this could lead to over-reporting.

Responses:

• The sentence has been corrected as suggested ,page 3

•The whole paragraph has been rephrased and some of the phrase have been removed. Page 3 of the clean manuscript

•More clarifications about the UN’s SDGs (goal 3 and 5) and how they relate to obstetric violence, has been added in page 3-4 of the clean manuscript

•The whole paragraph three, was rephrased and combined with the one talking about how culture and geographical factors influence obstetric violence

•These two paragraph have been rephrased and some of the phrased have been removed. Page 4-5

5 Material and Methods

•P. 4-5: Please add period after end of first paragraph.

• P. 5: First full paragraph: Please either add commas around “Tanzania being one of them,” or rearrange the first sentence.

•Also, consider providing additional support to the last sentence of this paragraph. This is a very important point, consider adding at least one more sentence that lays out the connection between obstetric violence and maternal mortality. Also please add a period at the end of this paragraph.

•Please explain the process for going from focus group and interviews into the creation of the prototype. What kind of analysis was done? By whom?

Response: The cited maternal mortality rate year has also been added page 5-6

•Additional information have been provided, showing the connection between obstetric violence and maternal mortality in page 5-6 and their periods

•More description about the process for going from focus group and interviews into the creation of the prototype and the kind of analysis that was done in the qualitative phase has been added in page 7-8

Phase III

Participants:

•P. 10- Please remove from first sentence: “if they are contextually valid.”

•P. 10- Consider adding the numbers of participants for each category of expert.

Response: The first sentence have been edited as suggested

•The number of participants for each category have been added in page 11

The workshop:

•P. 10- Please clarify the criteria the experts are using to rate the components. Were they rating whether or not these components should be included in the model? Or if these factors are important to their patients? Or which items they’ve had experience with? Please list the specific prompt.

P. 10- Please clarify how the experts provided feedback on the applicability of the component to the Tanzania context. Was this open-ended feedback?

Response:

•Some of the criteria used by the experts to rate the components of OV included: relevance, clarity and understandability, cultural sensitivity, applicability in a Tanzanian context, and fairness ensuring no bias toward any particular group.

•Each of the experts independently reviewed the components and provided written feedback. More information about this have been added to the manuscript in page 11-12

Phase V: Face validity testing

•P. 11- Same comment as above—please include the prompt the women were responding to on the Likert scale.

Response: Some of the prompts that postpartum women were responding to on the Likert scales included; the clarity of wording, simplicity of the language used and understand ability, cultural appropriateness and perceived value of each component .More description has been also added to the main manuscript in page 13

Analysis

•P. 13- Please clarify your criteria for considering recommendations with a “high prevalence.” How did you determine the cut-off for high-medium-low?

•P. 13- Please justify your use of 0.78 as your cut-off for the I-CVI and a cut-off of 0.75 for I-FVI.

Response:

• The criteria for considering a recommendation with high prevalence included the frequency of responses from experts, shared experiences among the experts, the significance of the component within the Tanzanian context and current evidence from other studies and reports referenced during the workshop. More description has been also added in page 14

• The cut-off points were determined based on expert judgment and consensus. A component was considered "high prevalence" if accepted by 75% or more of the experts, "medium prevalence" if accepted by 50%-74%, and "low prevalence" if accepted by less than 50% of the participating experts. This scoring system was also used by other studies. More description has been provided to the main manuscript in page 17

• Items with an I-CVI value of 0.78 or higher were considered to possess strong content validity that indicates that at least 78% of experts agreed that an item was relevant. Page 14- 15

• An I-FVI value of 0.75 or more was considered a good score signifies that at least 75% or more of respondents find the components clear and understandable. Page 15

6 Results

Maternal Health Experts:

•P. 14- The age group category in Table 2 is confusing as presented. Is there something you are trying to demonstrate with this variable? If this is meant to show that some of these groups are younger than others, there may be a more effective way to show that. If it is just to provide context, average age may be preferable.

Response: Table 2 has been amended and the mean age has been re-calculated. Page 16

Physical violence:

•P. 21- Please provide further explanation for “Kicking was reported as impractical.” Were these items assessed for practicality?

Response: Thank you for your thoughtful comment. After conducting a further review and additional consultations regarding “kicking to be impractical”, all authors have agreed that the item should be retained as part of the item on physical violence that may occur at both the community level and within facilities, as there have been reported cases of such incidents. The item has been added in table 4 page 19. The word impractical was mistakenly used

Lack of supportive care and treatment:

•P. 21- Please spell out what the acronym ANC stands for the first time it appears in the text.

Response: ANC stands for Antenatal Care ,has been spelled out in page 22

Sexual violence:

•Please provide further support for removing the category of rape. Provide additional support for this sentence: “The discussion led to the consensus that rape is not just an act of violence but a criminal offense, even though there have been reports of such actions by healthcare providers.” How did rape’s status as a crime fit into the discussion? What were the primary reasons laid out by the group for this item’s removal?

P. 23- Please clarify the sentence that begins with the phrase, “Some postnatal women…” Does this mean postnatal women suggested adding a new category: “to be forced to have sex with their partner during pregnant even when they are not feeling well”? Please explain, “claiming they are warm.”

Response:

•Thank you for your thoughtful review on the issue of rape.

• After consulting various legal frameworks, engaging with legal professionals within our context, and conducting further comprehensive literature review, all authors have reached a consensus to retain rape as a form of obstetric violence within the category of sexual violence as there have been reported cases of such incidents within our context .This classification applies when rape is perpetrated against a pregnant woman without her consent, whether in community settings or healthcare facilities . The rape item has been added in table 4 page 20 and more explanation with reference have been added in the discussion on page 27-28

• This was not suggested by postnatal women but during the face validation with them, they agreed this to be common and added more information to strengthen it .They said sometimes Partners may pressure them by asserting that sexual pleasure is enhanced during pregnancy

• This paragraph has also been rephrased for more clarity. Page 23-24

Psychological/emotional violence:

•P. 23- Please add the word “differently” to the following item: “treating a woman (differently) based on her age, parity and marital status…”

•P. 23- Please clarify this phrase, “experts and postnatal mothers added teenagers and high parity women as they are so vulnerable to violence during care provision at the health facilities in our settings.” Does this mean additional items based specifically on age and parity were added to the prototype?

Response: The word ‘differently’ have been added . Page 24

•Yes teenagers and high parity women were added to the prototype during the validation process, Based on experts experience it was agreed these two groups to be among the vulnerable groups for violence while pregnant and during labor. Page 24

7. Discussion

P. 26- Please clarify the sentence beginning with, “Experts also recommended the inclusion Health…” It is difficult to follow.

Response: “Health system conditions and constraints”, this category was derived from the World Health Organization (WHO) new evidence on mistreatment of women during childbirth. This document, was also one of the references used during the workshop. So during the plenary discussion experts suggested to be added as some of the listed items in this category, were also common in our settings. The whole paragraph has been rephrased too .Page 28

8. Strength and limitation

P. 27- Add a strength that highlights the practical utility of this OV prototype—what can it do, how can it advance the field?

Response: The strengths that highlights the practical utility of Ov in our settings ,have been added in page 28

P. 27- Please clarify this limitation. Is there any reason to assume that male partners and/or religious leaders will have more to say on OV than your current participants?

Response: Thank you for highlighting this important aspect of our research. The current participants offer invaluable first hand insights that significantly contribute to the depth of our findings. We also felt it was crucial to involve male partners and religious leaders in the validation process to ensure that their perspectives gathered during the qualitative phase, were properly represented. Their involvement could possibly allow them to assess the clarity of the language used, ensure that words were appropriately chosen , and incorporate any additional information that might not have been fully expressed during the interviews in the community . However, this was not a major concern, as we already had a strong representation of women and experts, who are also integral members of the community. So this limitation has been cancelled from the main manuscript.

•Consider including a limitation that addresses the difficulty of including all perspectives in one prototype for OV. How did you resolve scenarios of disagreement among your participants? How did you ensure that all of the voices in the focus groups were heard? Is it possible that relations of power played out in the focus groups, such that some voices were louder and therefore had outsized impacts on the resulting model?

Response: This limitation have been added in page 29

• When disagreements arose, researchers encouraged open dialogue by asking clarifying questions and prompting participants to elaborate on their viewpoints.

• Additionally, we established ground rules at the beginning of the workshop that emphasized respect for differing opinions.

• To ensure that all voices were heard in our focus groups, we utilized a cyclic format/rotating speaking during discussions, allowing each participant an equal opportunity to speak without interruption and to directly prompt quieter members to share their views, ensuring that no one was overlooked during the discussions.

• Furthermore, we provided anonymous feedback mechanisms such as written notes, so that quieter participants could share their thoughts without feeling pressured by dominant voices in the room.

9.Conclusion

•P. 27- Please lay out some of the more specific practical implications of your study—how can this be used to help pregnant people? To help healthcare providers? To address maternal mortality?

•Please lay out the next steps for researching this topic—what are the important questions that remain about OV in this context? What are the important areas for future research?

Response: More specific practical implications for this study such as t implications to pregnant women, health care providers and how it can be used to address maternal mortality have been added ni page 29-30

• Recommendations for future research and important areas to concentrate on have been added in page 30

Reviewer #2: Thank you for reviewing our manuscript we appreciate your comments

1. Abstract

Results: Seven categories of obstetric violence components were identified, including physical violence, lack of supportive care and treatment, subjugation care, Unfavorable care environment, sexual violence, verbal violence, emotional and psychological violence. In addition, 24 subcategories of obstetric violence were also identified .Issue: I suggest the removal of the space between identified and the full-stop

Response: This has been corrected

2. Funding details

The word doctoral is misspelled

Response: This has been corrected in page 32

3. Introduction:

Without acknowledging local expressions and interpretat

---

## [Decision Letter · Decision Letter 1]

PONE-D-24-51014R1Development and validation of context-specific components of obstetric violence: Experiences from Central Zone TanzaniaPLOS ONE

Dear Dr. Masoi,

Thank you for submitting your manuscript to PLOS ONE. After careful consideration, we feel that it has merit but does not fully meet PLOS ONE’s publication criteria as it currently stands. Therefore, we invite you to submit a revised version of the manuscript that addresses the points raised during the review process.

We look forward to receiving your revised manuscript.

Kind regards,

Florian Fischer

Academic Editor

PLOS ONE

Journal Requirements:

Reviewers' comments:

Reviewer's Responses to Questions

**Comments to the Author**

1. If the authors have adequately addressed your comments raised in a previous round of review and you feel that this manuscript is now acceptable for publication, you may indicate that here to bypass the “Comments to the Author” section, enter your conflict of interest statement in the “Confidential to Editor” section, and submit your "Accept" recommendation.

Reviewer #1: All comments have been addressed

Reviewer #2: All comments have been addressed

2. Is the manuscript technically sound, and do the data support the conclusions?

Reviewer #1: Yes

Reviewer #2: Yes

3. Has the statistical analysis been performed appropriately and rigorously? 

Reviewer #1: Yes

Reviewer #2: Yes

4. Have the authors made all data underlying the findings in their manuscript fully available?

Reviewer #1: Yes

Reviewer #2: Yes

5. Is the manuscript presented in an intelligible fashion and written in standard English?

Reviewer #1: Yes

Reviewer #2: Yes

6. Review Comments to the Author

Reviewer #1: This mixed-methods study investigates components of obstetric violence (OV) in the Central Zone of Tanzania, using postnatal mothers, healthcare providers and community members, ultimately identifying seven categories of OV, with acceptable levels of item-level content validity and item-face validity. All prior critiques and suggestions have been addressed appropriately. The methods are sound, and the revisions from the prior review have make this a strong contribution to the literature. Additional minor edits are suggested below.

Introduction:

• This work could still benefit from a careful revision to remove unnecessary (e.g., the dash in first sentence: “widespread issue-affecting”) and add needed (e,g., you are missing a space and a period here: “reported to be between44% and 52%, In Tanzania”) punctuation.

• P. 4- first sentence: please explain what you mean by “normalizing it.”

• P. 4- please add the word “national” to: “Tanzania’s national guidelines have outlined”

Materials and methods:

• P. 4- Please add a period an separate these into two sentences: “…experiences of obstetric violence at the facility level. As a result, some women avoid facility birth and then…”

Phase I: The Qualitative Study

• Please explain who “ten-cell leaders” are.

Verbal violence:

• P. 24- For this phrase: “especially if a woman fails to push the baby in a timely manner,” please add some context, such as “from the perspective of the healthcare provider.” You could indicate that healthcare providers have schedules/timelines in mind that have no correlation to the pregnant person’s health or wellbeing.

• P. 24- Also, please fix typo: “most common reason for why women hesitate to seek care”

Discussion:

• P. 26- fix typo: “components were valid and culturally acceptable”

• P. 26-27- Please consider adding one more sentence to the paragraph on the use of a syringe to manipulate nipples. What is the value of this exploration? Perhaps something that highlights the importance of collecting the full range of perspectives on these procedures in addition to collecting the medical communities’ perspective.

Reviewer #2: Dear Authors,

Congratulations on the work you have done so far, and I am glad to review it. My comments in the last round of review have been addressed. However, after going through the document, I noted that there are issues of grammar and punctuation that need to be addressed. If the issues that I have raised are addressed, then I believe this great work will be ready for publication. Issues that I have noted in my reading are indicated hereunder.

Further Comments

Introduction

Globally, estimates of obstetric violence range from 15% to 99% (3) with prevalence in Sub-Saharan Africa reported to be between44% and 52%, In Tanzania, obstetric violence remains a significant concern (4,5).  – Page 3.

Issue

Space needed between the word ‘between and 44%.

Despite ongoing research in examining the definition, root causes and methodologies for assessing obstetric violence occurring during pregnancy, childbirth and after childbirth, there is no universal consensus on how to define and measure obstetric violence (8).Cultural and… Page 3.

Issue

Space between the full stop at citation 8 and Cultural needed.

Tanzania’s guidelines have outlined the typology of disrespect and abuse in maternity care such as non-dignified care, abandonment, physical abuse, non-confidential care and non- consented care (10), yet they large draw from external context that may not fully reflect local realities (8,9,11).  Page 4

Issue

The word “large” should be replaced with the word “largely”

Comprehensive Emergency Obstetric and New-born Care (CEmONC) (13,14) Page 4.

Issue

Full stop needed at the end of citation 13 and 14.

Maternal health care provision remains a major challenge in developing countries such as Tanzania resulting in ongoing high maternal mortality (15) .One major risk factor… page 4

Issue

Full stop needed at the end of citation 15 needed and spacing before the word, “One”

Factors linked to home births in Tanzania encompass the hesitancy to seek care at a birthing facility due to individual and community experiences of obstetric violence at the facility level as a result , some women avoid facility birth and then experience preventable birth complications hence contributing to prevailing high maternal and neonatal mortality (17) .

Issue

Space between the word result and some to be deleted.

space between citation 17 and full stop to be removed. – check the rest of the citations for similar citation issues for the rest of the text.

This study was conducted in two regions in the central zone of Tanzania from July 2023 to June 2024 .

Issue

space between citation between 2024 and full stop to be removed.

It is the regions with -----

Issue

The word “regions” to be replaced with the word “region”

DISCUSSION

Obstetric violence is globally recognized as a serious issue threatening maternal health and

well-being, but there is no consensus on how best should be defined due to cultural and

geographical differences (18). 

Issue

The word “it” to be inserted between best and should…

Furthermore, feedback from maternal health experts, health care providers and postnatal mothers during the validation process were utilized and incorporated whereby some items were modified .

Issue

The space between modified and full stop to be removed These results align with a study conducted in the West Bank Palestine, where obstetric violence measurement tools were developed and validated, achieving a content validity index greater than 0.83 (41) indicating that all the components were valid and cultural acceptable .

Issue

The space between the word acceptable and full stop to be removed.

significant spiritual element of the childbirth experience (45) .

Issue

Space between citation 45 and the full stop to be removed.

There was particular concern about managing mentally distressed women during labor , considering staffing constraints.

Issue

Letter ‘a’ to be inserted between the words “was” and “particular”.

restraint was unnecessary in any situation and they perceived it as a display of healthcare providers lacking empathy (50) .

Issue

Spacing issues as highlighted in earlier comments.

… involves non-consensual sexual acts against pregnant women whether in community or healthcare settings (51–53)

Issue

Full stop needed after citation 51-53.

7. PLOS authors have the option to publish the peer review history of their article (what does this mean?). If published, this will include your full peer review and any attached files.

Reviewer #1: No

Reviewer #2: **Yes: **Melchizedek Nyakundi Mokaya

---

## [Author Response · Author response to Decision Letter 2]

26 May 2025

Reviewer #1: Thank you for reviewing our manuscript and for the constructive comment

1. Introduction

This work could still benefit from a careful revision to remove unnecessary (e.g., the dash in first sentence: “widespread issue-affecting”) and add needed (e,g., you are missing a space and a period here: “reported to be between44% and 52%, In Tanzania”) punctuation.

Response:

-Dear reviewer, thank you for your constructive comments

-The whole document have been revised more for grammatical errors and removing un necessary spaces and dash, we have also consulted the English language editor for proofreading and editing.

P. 4- first sentence: please explain what you mean by “normalizing it.”

Response:

-We appreciate your thoughtful comment.

-When we say, "normalizing obstetric violence, it means certain disrespectful childbirth practices become perceived as routine or even anticipated by some women that they will come across those acts as usual. This may occur due to repeated exposure, a lack of information about one's rights and a lack of adequate alternatives for access. As a result, some women normalize these practices as routine .We have also added more clarification in the manuscript in page 4

P. 4- please add the word “national” to: “Tanzania’s national guidelines have outlined

Response:

-The word “national” has been added .Page 4

2. Materials and methods

P. 4- Please add a period and separate these into two sentences: “…experiences of obstetric violence at the facility level. As a result, some women avoid facility birth and then…

Response:

-The sentence has been revised for clarity. Page 5

Phase I: The Qualitative Study

• Please explain who “ten-cell leaders” are.

Response:

-Ten-cell leaders are grassroots administrative leaders in Tanzania who build close relationships with household. Ten-cell leaders' understanding of family relationships, domestic issues and the local government, positions them to be knowledgeable and speak about their community.

-We have also added clarification in the manuscript in page 7

3. Results

Verbal violence:

P. 24- For this phrase: “especially if a woman fails to push the baby in a timely manner,” please add some context, such as “from the perspective of the healthcare provider.” You could indicate that healthcare providers have schedules/timelines in mind that have no correlation to the pregnant person’s health or wellbeing.

P. 24- Also, please fix typo: “most common reason for why women hesitate to seek care”

Response:

-Thank you for this insight

-We have added more clarification on this paragraph as suggested and the typo have been fixed. Page 25

4. Discussion

P. 26- fix typo: “components were valid and culturally acceptable”

P. 26-27- Please consider adding one more sentence to the paragraph on the use of a syringe to manipulate nipples. What is the value of this exploration? Perhaps something that highlights the importance of collecting the full range of perspectives on these procedures in addition to collecting the medical communities’ perspective

Response:

-We appreciate this useful suggestion. We agree that highlighting the need to capture a diversity of perspectives enhances the discussion.

- An additional sentence has been included to bring out the need to learn about how such procedures are perceived not only by healthcare providers but also by the women who experience them and which is significant in informing respectful and culturally sensitive maternal care practices. Page 28

Reviewer #2: Thank you for reviewing our manuscript we appreciate your feedback

-Congratulations on the work you have done so far, and I am glad to review it .

-My comments in the last round of review have been addressed. However, after going through the document, I noted that there are issues of grammar and punctuation that need to be addressed. If the issues that I have raised are addressed, then I believe this great work will be ready for publication. Issues that I have noted in my reading are indicated hereunder

Response:

-Dear reviewer ,we appreciate this recognition and insightful feedback

-We have gone through the whole document and work on the grammatical issues and punctuation with the help of English language editor for proofreading and editing

1. Introduction

Globally, estimates of obstetric violence range from 15% to 99% (3) with prevalence in Sub-Saharan Africa reported to be between44% and 52%, In Tanzania, obstetric violence remains a significant concern (4,5). – Page 3.

Issue

Space needed between the word ‘between and 44%

Response:

-The space has been added . Page 3

Despite ongoing research in examining the definition, root causes and methodologies for assessing obstetric violence occurring during pregnancy, childbirth and after childbirth, there is no universal consensus on how to define and measure obstetric violence (8).Cultural and… Page 3.

Issue

Space between the full stop at citation 8 and Cultural needed.

Response:

-The space has been added. Page 4

Tanzania’s guidelines have outlined the typology of disrespect and abuse in maternity care such as non-dignified care, abandonment, physical abuse, non-confidential care and non- consented care (10), yet they large draw from external context that may not fully reflect local realities (8,9,11). Page 4

Issue

The word “large” should be replaced with the word “largely”

Response:

-The word “large” has been edited . Page 4

Comprehensive Emergency Obstetric and New-born Care (CEmONC) (13,14) Page 4.

Issue

Full stop needed at the end of citation 13 and 14.

Response:

-A full stop has been added. Page 4

Maternal health care provision remains a major challenge in developing countries such as Tanzania resulting in ongoing high maternal mortality (15) .One major risk factor… page 4

Issue

Full stop needed at the end of citation 15 needed and spacing before the word, “One”

Response:

-A full stop and space have been added. Page 4

Factors linked to home births in Tanzania encompass the hesitancy to seek care at a birthing facility due to individual and community experiences of obstetric violence at the facility level as a result , some women avoid facility birth and then experience preventable birth complications hence contributing to prevailing high maternal and neonatal mortality (17) .

Issue

Space between the word result and some to be deleted.space between citation 17 and full stop to be removed. – check the rest of the citations for similar citation issues for the rest of the text.

Response:

-The space has been removed . Page 5

This study was conducted in two regions in the central zone of Tanzania from July 2023 to June 2024 .

Issue

space between citation between 2024 and full stop to be removed.

Response:

-The space has been removed

2. DISCUSSION

Obstetric violence is globally recognized as a serious issue threatening maternal health and well-being, but there is no consensus on how best should be defined due to cultural and geographical differences (18).

Issue

The word “it” to be inserted between best and should…

Response:

-The sentence has been rectified . Page 26

Furthermore, feedback from maternal health experts, health care providers and postnatal mothers during the validation process were utilized and incorporated whereby some items were modified

Issue

The space between modified and full stop to be removed

Response:

-The sentence has been revised and space removed . Page 26

These results align with a study conducted in the West Bank Palestine, where obstetric violence measurement tools were developed and validated, achieving a content validity index greater than 0.83 (41) indicating that all the components were valid and cultural acceptable .

Issue

The space between the word acceptable and full stop to be removed

Response:

-The space has been removed as seen in page 27

Significant spiritual element of the childbirth experience (45).

Issue

Space between citation 45 and the full stop to be removed

Response:

-The space has been removed

There was particular concern about managing mentally distressed women during labor, considering staffing constraints.

Issue

Letter ‘a’ to be inserted between the words “was” and “particular

Response:

-The letter ‘a’ has been inserted . Page 28

… involves non-consensual sexual acts against pregnant women whether in community or healthcare settings (51–53)

Issue

Full stop needed after citation 51-53.

Response:

-A full stop has been inserted. Page 29

---

## [Editor Report · Decision Letter 2]

Development and validation of context-specific components of obstetric violence: Experiences from the Central Zone of Tanzania

PONE-D-24-51014R2

Dear Dr. Masoi,

We’re pleased to inform you that your manuscript has been judged scientifically suitable for publication and will be formally accepted for publication once it meets all outstanding technical requirements.

Kind regards,

Florian Fischer

Academic Editor

PLOS ONE
---

## [Editor Report · Acceptance letter]

PONE-D-24-51014R2

PLOS ONE

Dear Dr. Masoi,

I'm pleased to inform you that your manuscript has been deemed suitable for publication in PLOS ONE. Congratulations! Your manuscript is now being handed over to our production team.

Kind regards,

on behalf of

Dr. Florian Fischer

Academic Editor

PLOS ONE